# Australian University Nursing and Allied Health Students’ and Staff Physical Activity Promotion Preparedness and Knowledge: A Pre-Post Study Using an Educational Intervention

**DOI:** 10.3390/ijerph19159255

**Published:** 2022-07-28

**Authors:** Nicole Freene, Katie Porra, Jaquelin A. Bousie, Mark Naunton, Nick Ball, Andrew Flood, Kasia Bail, Sally De-Vitry Smith, Milli Blenkin, Lynn Cheong, Madeleine Shanahan, Stephen Isbel, Myra Leung, Ann B. Gates

**Affiliations:** 1Physiotherapy, Faculty of Health, University of Canberra, Bruce, ACT 2617, Australia; u3204570@uni.canberra.edu.au (K.P.); jaquelin.bousie@canberra.edu.au (J.A.B.); 2Health Research Institute, University of Canberra, Bruce, ACT 2617, Australia; 3Pharmacy, Faculty of Health, University of Canberra, Bruce, ACT 2617, Australia; mark.naunton@canberra.edu.au (M.N.); lynn.cheong@canberra.edu.au (L.C.); 4Sport and Exercise Science, Faculty of Health, University of Canberra, Bruce, ACT 2617, Australia; nick.ball@canberra.edu.au; 5Psychology, Faculty of Health, University of Canberra, Bruce, ACT 2617, Australia; andrew.flood@canberra.edu.au; 6Research Institute for Sport and Exercise, University of Canberra, Bruce, ACT 2617, Australia; 7Nursing, Faculty of Health, University of Canberra, Bruce, ACT 2617, Australia; kasia.bail@canberra.edu.au; 8Midwifery, Faculty of Health, University of Canberra, Bruce, ACT 2617, Australia; sally.de-vitrysmith@canberra.edu.au; 9Counselling, Faculty of Health, University of Canberra, Bruce, ACT 2617, Australia; milli.blenkin@canberra.edu.au; 10Medical Radiation Science, Faculty of Health, University of Canberra, Bruce, ACT 2617, Australia; madeleine.shanahan@canberra.edu.au; 11Occupational Therapy, Faculty of Health, University of Canberra, Bruce, ACT 2617, Australia; stephen.isbel@canberra.edu.au; 12Vision Science & Optometry, Faculty of Health, University of Canberra, Bruce, ACT 2617, Australia; myra.leung@canberra.edu.au; 13Faculty of Medicine and Health, University of Nottingham, Nottingham NG7 2RD, UK; annbgates@googlemail.com; 14Faculty of Medicine and Health, Plymouth Marjon University, Plymouth PL6 8BH, UK

**Keywords:** physical activity, teaching, education, curriculum, health, evaluation

## Abstract

The promotion of physical activity (PA) by health professionals is a key strategy to increase PA levels in the population. In this study, we investigated PA promotion, preparedness, and knowledge among university nursing and allied health students and staff, as well as PA resource usage within curricula, before and after an educational intervention. Students and staff from 13 health disciplines at one Australian university were invited to complete an online survey, and a curriculum audits were conducted before and after PA teaching resources were promoted by academic PA champions (*n* = 14). A total of 299 students and 43 staff responded to the survey pre-intervention, and 363 and 32 responded to the post-intervention, respectively. PA promotion role perception (≥93%) and confidence to provide general PA advice (≥70%) were high throughout the study. Knowledge of PA guidelines was poor (3–10%). Students of physiotherapy, sport and exercise science, as well as more active students, were more likely to be aware of the PA guidelines (*p* < 0.05). Over 12 months, PA promotion preparedness and knowledge did not change significantly, nor was there a change in the amount of PA content delivered, despite a significant increase in the use of the teaching resources across a number of disciplines (*p* = 0.007). Future research should be carried out to investigate the implementation of the resources over time and to develop additional strategies for PA promotion and education scaffolded across curricula.

## 1. Introduction

Insufficient physical activity (PA) is a major risk factor for chronic disease and death, and has recently been identified as a risk factor for severe illness resulting from COVID-19 [1,2]. It is estimated to cost Australia, the country in which this study was conducted, AUD 805 million annually [3]. Currently, 27% of Australian adults are not meeting the national PA guidelines [4]. PA promotion by health professionals is viewed as a key strategy to increase the PA levels in the population. This strategy has been outlined in the ‘*World Health Organization (WHO) Global Action Plan on Physical Activity*’ (GAPPA) and the ‘*Blueprint for an active Australia*’ [5,6]. There is evidence that even brief PA interventions provided by health professionals, such as brief PA advice, are just as effective as more intensive interventions, such as exercise referral schemes [7,8]. Health professionals agree that providing PA advice is part of their role, and they have a broad reach within the population, with 21 million Australians consulting at least one health professional every year [9,10]. Despite this, only 16% of Australian health professionals are able to describe the PA guidelines accurately [9].

The target of the WHO GAPPA is a 15% reduction in the global prevalence of physical inactivity in adults and adolescents by 2030 [6]. Action items within GAPPA convey the need to strengthen the pre- and post-service training of health professionals to increase their knowledge and skills in order to encourage an active society (action 1.4) and implement and strengthen patient assessment and counselling in relation to increasing PA and reducing sedentary behaviour (action 3.2) in order to reach this target. Gates et al. (2020) suggest that undergraduate health professional training related to PA, as well as examining and assessing PA knowledge and skills within higher education institutions, are crucial to implementing the GAPPA’s objectives [11]. Assessment of this training is essential as it is used to measure student learning in the process of attaining graduate profiles, thus reinforcing the message that this aspect of the curriculum is important and ensuring that graduates master critical academic and professional outcomes [12,13]. Additionally, there is evidence that awareness and knowledge of the PA guidelines is associated with increased PA promotion by health professionals [9,14].

Few studies have investigated educational interventions relating to PA promotion for nursing and allied health students. Studies have been conducted with nursing [15,16], pharmacy [17,18,19], physiotherapy [20], and sport and exercise students [21], as well as mixed-health students [22], with wide variations in the educational interventions’ duration (ranging from 1 h [22] to 3 years [20]), delivery, and content. Confidence in promoting PA, attitudes, knowledge, role perception, and personal PA levels have been assessed and there is some evidence that PA education can be effective in improving these outcomes [15,16,17,18,19,21,22]. However, the quality of these studies is limited, with no randomised control groups and small sample sizes. Attempts have been made to create a consensus on the PA categories and topics to be included in health professional training [23]. Nevertheless, current academic training on PA promotion within nursing and allied health disciplines appears to be inconsistent and lacking [23]. Therefore, the aim of this study was to (i) evaluate the PA promotion preparedness and knowledge of Australian university nursing and allied health students and staff; and (ii) determine the use of PA resources within health curricula before and after the promotion of an international PA teaching resource among the academic staff of 13 nursing and allied health disciplines at one Australian university (Appendix A).

## 2. Materials and Methods

### 2.1. Study Design

Online surveys and curriculum audits were conducted between August and November 2020 (T1) and 2021 (T2), before and after the intervention. Using voluntary sampling, University of Canberra Faculty of Health students and staff from 13 health disciplines (counselling, psychology, physiotherapy, pharmacy, medical radiation science, vision science and optometry, occupational therapy, sport and exercise science (including exercise physiology), public health, speech pathology, nutrition and dietetics, nursing, and midwifery) were recruited through advertisements placed on the online study platform and via email, containing a link to the online survey. Two reminder advertisements were sent to students and staff during the data collection periods. Student survey participants were included if they self-reported that they were a student in the Faculty of Health and were planning a career as a health professional. Staff survey and curriculum audit participants were included if they self-reported that they were an academic staff member in the Faculty of Health. No incentives were provided for completing the survey.

### 2.2. Intervention

#### 2.2.1. Movement for Movement Physical Activity Resources

The Movement for Movement resources are a worldwide interdisciplinary, pre-qualification teaching resource on exercise medicine for the prevention and treatment of non-communicable diseases [24,25]. The resources are free, requiring permission for use, and are updated frequently by international PA, education, and health professional experts, including health students, and have been designed and evaluated specifically for undergraduate health care professional use. The PA resources consist of national and international strategies and infographics, background introductions, specific disease and health condition slide-sets, a text module, and advice on how to use the resources effectively. The WHO has highlighted the Movement for Movement resources as an example for creating change and has applauded their high quality [26]. The resources are currently being used in schools of nursing, physiotherapy, medicine, and pharmacy around the world [27].

#### 2.2.2. Physical Activity Champions

Discipline leaders identified PA champions within their disciplines (*n* = 14). PA champions were academic teaching staff from each health discipline who volunteered to assist with promotion of the Movement for Movement resources and the recruitment of participants, both students and staff, within their discipline. PA champions were initially trained on the use of the resources via an online workshop (AG). The research team regularly contacted the PA champions via email and in person to provide support and recommendations for using the resources, including the need for assessment. The PA champions worked closely with their discipline to encourage the use and implementation of these resources into existing and new subjects. A flexible approach was utilised, with no compulsory use of the resources and no standardisation of how these resources were implemented, for example, the year level delivered, the number of lectures/tutorials, whether the materials were delivered in-person or via self-directed learning online, or assessment methods. There were also no restrictions placed on the use of other PA resources or teaching staff creating their own PA content, with the overall aim of increasing the amount of PA content delivered across all disciplines within the 12 month intervention period (December 2020–November 2021).

### 2.3. Survey

The online survey, implemented using the Qualtrics platform (Qualtrics, Provo, UT, USA), was based on a previous questionnaire that has been used in nursing and allied health professions, including physiotherapy students, to assess PA promotion preparedness and knowledge [9,28]. The survey was adapted to suit students and staff in all disciplines by modifying ‘patients’ to ‘patients/individuals’ and using ‘in your discipline’, to increase the specificity and relevance of responses. The survey was designed to take approximately 10 min to complete.

Participants were asked about their PA promotion preparedness (their perceived role in PA promotion, barriers to promotion, the feasibility of different promotion strategies, personal PA levels) and their knowledge of the Australian Physical Activity and Sedentary Behaviour Guidelines (the guidelines) for adults (18–64 years). Most items of the survey were scored using a five-point Likert scale, for example, strongly agree to strongly disagree. The guideline responses were assessed for total completion, as well as the acceptability of the descriptions provided for the four components of duration, intensity, strength/resistance training, and minimising sedentary behaviour. A description was considered acceptable if it broadly included a duration of 150–300 min of moderate-intensity or 75–150 min of vigorous-intensity PA or a combination of both, strength/resistance training twice a week, and minimising sedentary behaviour and/or reducing prolonged sitting periods.

The surveys included general demographic questions, on factors such as age, gender, and discipline. The student survey also asked for the participant’s year of university study, and whether they were an undergraduate or postgraduate student (Appendix A). The staff survey asked whether the participant was a trained health professional and if they were currently practising clinically. Those who were practising clinically were asked further questions, including the type of place in which they worked, the number of years since they completed their first health degree, number of patients seen each week, the number of hours worked each week, and how often they promoted a physically active lifestyle to their patients (Appendix A).

### 2.4. Curriculum Audit

The curriculum audit was conducted to determine the use of PA resources within health curricula. PA champions contacted staff in their discipline to determine which subjects had PA content focusing on health promotion and preventive health. Subjects which included content on therapeutic exercise only—defined as the prescription of an exercise program that involved the patient undertaking voluntary muscle contraction and/or body movement with the aim of relieving symptoms; improving function; and improving, retaining, or slowing the deterioration of health [29]—were excluded from the audit. Emails were sent to the identified subject convenors with the auditing tool and a brief description of what was required. Staff had the option to complete the tool themselves or to meet with the research team for <15 min to complete the document. The audit recorded details for each subject, including the year of their degree, the number of students, whether the PA content was presented in lectures or tutorials, how much time was spent on this content, and whether the PA content was assessed (Appendix A). Details also considered whether age-specific guidelines were used in each subject, whether the guidelines were explicitly stated including all four components, and whether the subject included any health benefits of PA and/or PA promotion methods, such as techniques relatingto behavioural changes.

### 2.5. Sample Size

The sample sizes used for the student and staff survey were based on the total number of students enrolled (*n* = 3931) and academic staff employed (*n* = 152) within the Faculty of Health in semester 2 (August–November) 2020. Using an online survey sample size calculator, with a confidence interval of 95% and a 5% margin of error, the sample sizes required for the students and staff were 350 and 109, respectively.

### 2.6. Data Analysis

All returned surveys were included in the analysis. Responses were categorised according to the established guidelines (complete: 80–100% of questions answered; partially complete: 50–79% of questions answered; incomplete: <50% of questions answered) [30]. Data were analysed using descriptive analyses, including frequencies, percentages, means and standard deviations, medians, and inter-quartile ranges. To facilitate analysis and reporting, each of the Likert scale questions were transformed into binary variables (i.e., agree/often versus neutral/disagree). The normality of continuous data was assessed using the Kolmogorov–Smirnov test unless the sample size was <50, in which case the Shapiro–Wilk test was used. For the pre- and post-intervention comparison, independent *t*-tests or Mann–Whitney U tests were used, where appropriate. Chi-squared tests of independence were performed for categorical data. In cases where cells had counts of less than five, Fisher’s exact test was used. Chi-squared tests of independence and Pearson correlations were used to assess associations between categorical variables. For the awareness and knowledge of the guidelines, only fully completed surveys were included for students and staff, assuming no answer for the open text knowledge question, which indicated no knowledge of the guidelines. All quantitative analyses were conducted using the Statistical Package for Social Sciences (SPSS), version 27 (IBM Corp., Armonk, NY, USA). The significance level was set at *p*  ≤  0.05. Open text comments from the survey were analysed using Braun and Clarke’s (2006) thematic analysis approach [31]. Two experienced qualitative research team members independently coded and identified themes (NF, JB). The emerging themes were discussed until a consensus was reached on final themes and categories.

## 3. Results

At baseline, 299 eligible students completed the survey, with 267 students (89%) retuning a complete survey. The median age of students was 24 years and the majority were female (Table 1). Over a quarter of the students were studying nursing and most were undergraduates, in the first year of their degree. Post-intervention, 363 eligible students completed the survey, with 329 students (91%) returning a complete survey. Comparing student participants pre- and post-intervention, there were no differences in characteristics, except for discipline (*p* = 0.008).

Forty-three staff completed the survey at baseline, with 38 (88%) returning a complete survey. The majority of participants were female and half were aged ≥45 years (Table 1). Nearly a quarter of staff were from the discipline of nursing and most staff were trained health professionals. There was an average of 22 years since staff completed their first degree in health. Of the 32 participants who were trained health professionals, half were currently practicing clinically. Approximately half of the practicing clinical staff encouraged six or more patients a month to have a more physically active lifestyle. The most common place of work was private practice. These staff saw an average of nine patients per week and worked approximately 13 clinical hours each week. The average number of years in practice for currently practising clinical staff was 19 years. Post-intervention, 32 eligible staff completed the survey, with 30 staff (94%) returning a complete survey. There were no differences in staff participant characteristics pre- and post-intervention.

### 3.1. Knowledge, Role Perception, Confidence, Barriers, and Feasibility in Regard to Physical Activity Promotion

Half of the students and less than half of the staff agreed with the public health PA messages that 30 min/day and generally being more active everyday was good for health (Table 2). Only one third of students and staff agreed that moderate-intensity PA (‘puff and pant’) is good for health. Nearly all students and staff (≥93%) agreed that it is part of the health professional’s role to discuss the benefits of a physically active lifestyle with patients/individuals. Nearly all students and staff thought that health professionals should be physically active to act as role models and that they should suggest ways for patients/individuals to increase their PA. Three quarters of students and staff felt confident in giving general PA advice, whereas approximately half felt confident suggesting specific PA programs. Lack of time and counselling skills were perceived to be the biggest barriers to PA promotion, with approximately 15% of students and staff feeling that promoting PA would not be of benefit for the patient/individual. Brief counselling integrated into regular consultations was perceived to be the most feasible PA promotion strategy.

For students (*p* = 0.05) and staff (*p* = 0.02), there was a significant increase in agreement with the public health message “Several short walks on most days is better than one round of golf per week for good health” from pre- to post-intervention (Table 2). There were no other significant changes in knowledge of the PA messages, role perception, confidence, barriers, or feasibility of PA promotion strategies.

### 3.2. Other Barriers to Promoting a Physically Active Lifestyle to Patients/Individuals

Sixty-eight students and 12 staff provided open-text responses on other barriers to PA promotion. The qualitative analysis revealed four themes, which were common for students and staff (Table 3). Lack of PA promotion skills: Participants indicated a lack of confidence and skills in promoting PA. This included both the promotion of general and specific PA. Some participants were also concerned about addressing this behaviour with patients, feeling that it might cause offence and damage rapport. Feeling hypocritical: A number of participants felt that they could not promote something they were not doing themselves, reporting feelings of hypocrisy if they themselves were inactive. Understanding the benefits of PA promotion by all HPs, for all patients, in all settings: Participants reported that other issues and treatments often took priority, and for some disciplines this was outside of their role as a health professional, particularly in an acute setting. Organisational support was recognised as key: Several participants wrote that organisational support and reinforcement was necessary for PA promotion by health professionals. Key factors raised were workload considerations and that promotion within their workplace was needed to support this intervention.

### 3.3. Awareness and Knowledge of the Australian Physical Activity and Sedentary Behaviour Guidelines

At baseline, less than half of the students and staff were aware of the guidelines. Only 3% (*n* = 7/267) of students and 5% (*n* = 2/38) of staff were able to accurately describe all four components of the guidelines (duration, intensity, strength training, sedentary behaviour; Appendix A). Post-intervention, there was no significant change in awareness and knowledge of the guidelines, although there was a non-significant increase in awareness amongst staff (Figure 1; Appendix A).

For students, there were no associations between awareness or knowledge of the guidelines and age, gender, or undergraduate versus postgraduate degrees, pre- or post-intervention. Pre-intervention, there was no association between year level and awareness, although post-intervention this association was significant (X^2^(3) = 13.7, *p* = 0.003), with awareness increasing with increasing year levels. Both pre- (X^2^(3) = 9.4, *p* = 0.01) and post-intervention (X^2^(3) = 14.3, *p* = 0.001), there was a significant association between year level and knowledge, with knowledge increasing with increasing year levels. Pre-intervention, there was no association between discipline or personal PA levels and knowledge, although post-intervention these associations were significant (discipline: X^2^(12) = 34.6, *p* < 0.001; personal PA: X^2^(4) = 16.4, *p* < 0.001). Physiotherapy, sport and exercise science and students who were much more active were more likely to know the guidelines post-intervention. Both pre- (X^2^(12) = 51.5, *p* = 0.001) and post-intervention (X^2^(12) = 90.7, *p* = 0.001), there was a significant association between discipline and awareness of the guidelines. Physiotherapy and sport and exercise science students were more likely to be aware. There was also a significant association between awareness of the guidelines and personal PA levels pre- (X^2^(4) = 11.1, *p* = 0.025) and post-intervention (X^2^(4) = 14.3, *p* = 0.006), with more active students more likely to be aware of them.

For staff, there were no associations between awareness or knowledge of the guidelines and age, gender, trained health professional, years since degree, practicing health professional, practice type, PA promotion frequency, number of patients/wk, years in practice, and clinical hours/wk. There were no associations between discipline or PA levels and knowledge of the guidelines pre- and post-intervention. Pre-intervention, there was a significant association between both discipline (X^2^(10) = 17.9, *p* = 0.01) and PA levels ((X^2^(4) = 13.2, *p* = 0.005) and awareness of the guidelines; however, these associations were no longer significant post-intervention. Participants from five disciplines were not aware of the guidelines pre-intervention and participants from only one discipline were not aware post-intervention. Staff self-reporting higher PA levels were more likely to be aware of the guidelines pre-intervention only.

### 3.4. Curriculum Audit

At baseline, 41 subjects as reported by the faculty contained PA content in their curricula, across eight disciplines (Table 4). Physiotherapy, sport and exercise science, and midwifery had the most subjects containing PA content. PA content was most commonly delivered as a first-year undergraduate subject, taking one hour to deliver within a combination of lectures and tutorials. The duration of PA content increased to two hours post-intervention and the number of subjects including PA content increased to 48, covering all disciplines except counselling and speech pathology, but these changes were not significant. Only half of the subjects explicitly outlined the guidelines and approximately half the subjects assessed the PA content. Over half (63–75%) of the subjects covered PA promotion strategies. Overall, there were no significant changes in the amount of PA content within the Faculty of Health between 2020 and 2021. However, there was a significant increase in the number of subjects using the Movement for Movement resources with almost 30% of subjects in the curricula audit using these resources in 2021 compared to just 5% in 2020 (*p* = 0.007).

## 4. Discussion

The availability of international PA teaching resources, promoted by academic PA champions, resulted in no significant change in student or staff PA guideline awareness and knowledge, which remained poor, and no significant change in the amount of PA content delivered in pre-qualification nursing and allied health subjects over 12 months. However, there was a significant increase in the number of subjects using the peer-reviewed PA curriculum and an increase in agreement with general public health PA messaging, among both students and staff. Similarly to practicing health professionals in Australia [9], a large majority of students and staff from 13 nursing and allied health disciplines agreed that discussing the benefits of PA with their patients was part of the health professional’s role, and most felt confident providing general PA advice throughout the data collection period. Regardless of this, students’ discipline and personal PA levels were associated with guideline awareness and knowledge, which may impact PA promotion post-qualification. Utilising a package of PA teaching resources and PA champions may assist with embedding a PA curriculum within academic nursing and allied health disciplines, potentially leading to changes in the PA promotion skills of pre-qualification health students, but time may be required for implementation and other strategies should be considered to improve PA promotion skills.

Previous studies using educational interventions to increase the PA promotion preparedness of nursing and allied health students reported good levels of success [15,16,17,18,19,21,22]. In contrast to our study, outcomes were measured at the beginning and end of a specific intervention in a specific cohort, for example, pre- and post-subject surveys were used, with no longer-term follow-up (>6 months). Here we measured outcomes at the same time-point, 12 months apart, across all subjects, year levels, and disciplines. Only one study implemented a discipline specific intervention across the entire curriculum and assessed the personal PA frequency of students 6 years apart, finding a non-significant increase [20]. The lack of significant improvements observed in our study may be due to the timing of data collection, with outcomes not being collected immediately after students had received the PA promotion education. Additionally, some outcomes were already highly positive at baseline, for example, role perception and confidence, with little room for improvement.

Key PA categories and topics, and steps to embed PA promotion into health professional curriculums have been identified [23,32]. Using an e-Delphi process with 73 experts from seven health professions, health behavioural change was identified as the most important category to include in PA health professional training, followed by clinical exercise physiology (PA benefits, assessment, and prescription), and PA and public health (knowledge of guidelines, programs, and interventions) [23]. These topics were deemed crucial to include in health professional training, including the role each discipline can play, and the authors reported that they should be scaffolded throughout health curricula. Milton et al. (2020) provided a more specific outline of how PA can be embedded within health curricula, identifying five areas of action, including the creation and provision of resources with support [33]. Our use of the Movement for Movement resources and PA champions was in alignment with these three action areas. The two remaining action areas, the inclusion of PA within subject learning outcomes and PA promotion problem-based learning scenarios, were not specifically included in our study. Although these elements may have been included in some of our subjects, there was not a consistent approach across or within disciplines. The inclusion of PA within learning outcomes ensures that PA content is taught. The use of problem-based learning scenarios, case studies, simulated consultations, and clinical placement opportunities enables students to apply their PA promotion skills and these learning activities have been found to be beneficial in medical training [34]. Assessment is also key, as without the examination of PA promotion knowledge and skills, students will not consider this a core competency [35].

The qualitative themes indicated that students and staff wanted to know how to promote PA, and identified a need to understand that this can be carried out by any health professional. This reinforces the above recommendations regarding what should be incorporated into health professional curriculums and how, for example, to utilise practical opportunities. The themes also highlighted a need for interprofessional learning to improve these skills and the understanding of the role that each profession can play. Students and staff also reported an association between personal PA levels and PA promotion, suggesting that a lack of personal PA evoked a feeling of being hypocritical. This association was supported by the survey results, showing that active students were more likely to be aware and to have knowledge of the guidelines. Active staff were also more likely to be aware of the guidelines pre-intervention. Similarly, systematic reviews have shown that active health professionals are more likely to promote PA [14,36]. One strategy to increase PA promotion skills may be to include learning activities that address health student’s personal PA levels. Additional strategies may include PA promotion training for staff and providing a more structured approach for the implementation of the resources. Implementation is likely to be highly individualised due to already full curricula and external requirements for course accreditation, with health promotion and preventive health potentially not being a focus, although this is less likely in exercise science and physiotherapy courses. Thus, further research is required to determine the most effective methods for the implementation of PA promotion training into existing curriculums. The use of PA champions may be particularly important in order to influence this implementation process, mapping this content within disciplines and encouraging a variety of methods to embed PA within curricula. Time for implementation also appears to be necessary, with the collection of outcome measures over the longer term recommended in order to evaluate changes in health students’ PA preparedness and knowledge.

There are a number of strengths and limitations of this study. This study was conducted with health staff and students from several health disciplines where, to our knowledge, PA promotion preparedness and knowledge before and after an educational intervention had not been previously evaluated. However, this study was only conducted at one university in Australia and results may not be generalisable to other university settings. Our student survey response rate was high, reaching the required sample size post-intervention. Selection bias may have occurred due to voluntary sampling, with students and staff more interested in PA potentially being more likely to complete the survey or less likely to complete the survey as they felt that this was their core business (sport and exercise science, physiotherapy). Social desirability bias may have also been present, for example, participants may have over-estimated their personal PA levels. There is also the possibility that some participants were not University of Canberra Faculty of Health students or staff, and were not planning a career as a health professional, as the online survey was self-administered. Additionally, the promotion of the PA resources may not have been consistent amongst all disciplines and the COVID-19 pandemic may have affected their implementation. In 2021, the university altered teaching methods due to social distancing requirements, encouraging online teaching where possible. During the second semester (August–December 2021), the university campus was closed due to COVID-19, forcing all subjects online and curriculums to be re-shuffled to accommodate this change in delivery. Finally, the use of the independent *t*-test and Chi-squared test for the pre/post comparison may not have been appropriate as it is unclear if there was an independence of observations in all cases.

## 5. Conclusions

It is possible to embed international PA teaching resources across a number of health disciplines via the promotion of the PA resource and the effort of academic PA champions. However, in 12 months, this did not result in significant improvements in PA promotion preparedness or knowledge for students or staff, nor did it result in a significant increase in the amount of PA content delivered, which was low. Additional strategies for PA promotion and education within nursing and allied health degrees need to be considered, including a variety of educational approaches, such as PA promotion behavioural change practice and increasing students’ personal PA levels. These learning activities should be scaffolded across degrees, along with the assessment of the skills necessary to ensure their mastery. Preparing our future health workforce to take advantage of PA promotion opportunities must be a priority in order to improve the health and well-being of their future patients.

## Figures and Tables

**Figure 1 ijerph-19-09255-f001:**
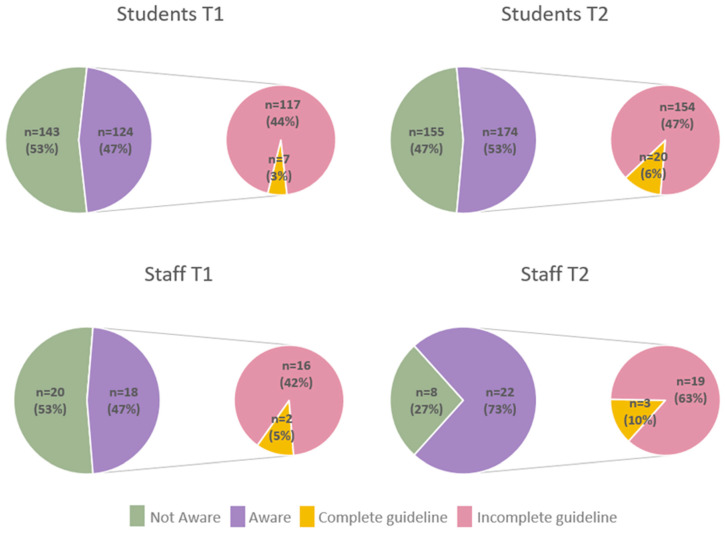
Awareness and knowledge of the Australian Physical Activity and Sedentary Behaviour guidelines among health students and staff pre- (T1) and post-intervention (T2).

**Table 1 ijerph-19-09255-t001:** Student and staff participant characteristics pre- and post-intervention.

	Student		Staff	
Characteristic	T1	T2	T1	T2
	(*n* = 267)	(*n* = 330)	(*n* = 38)	(*n* = 30)
Gender, Female *n* (%)	230 (86.1)	271 (82.1)	28 (73.7)	23 (76.7)
Student age (yrs), median (Q1, Q3)	24 (21, 33)	23 (20, 31)	-	-
Staff age (yrs), ≥45 yrs *n* (%)	-	-	19 (50)	16 (53.4)
Discipline, *n* (%)				
Occupational Therapy	21 (7.9)	11 (3.3) *	3 (7.9)	1 (3.3)
Physiotherapy	41 (15.4)	56 (17.0)	5 (13.2)	8 (26.7)
Pharmacy	24 (9)	12 (3.6)	1 (2.6)	1 (3.3)
Nutrition and Dietetics	12 (4.5)	15 (4.5)	-	-
Nursing	79 (29.7)	122 (37.0)	9 (23.7)	11 (36.7)
Midwifery	13 (4.9)	12 (3.6)	6 (15.8)	4 (13.3)
Psychology	41 (15.4)	40 (12.1)	4 (10.5)	1 (3.3)
Vision Science and Optometry	3 (1.1)	9 (2.7)	2 (5.3)	1 (3.3)
Speech Pathology	4 (1.5)	7 (2.1)	-	-
Counselling	2 (0.8)	3 (0.9)	3 (7.9)	1 (3.3)
Sport and Exercise Science	11 (4.1)	30 (9.1)	2 (5.3)	2 (6.7)
Medical Radiation Science	6 (2.3)	8 (2.4)	-	-
Public Health	9 (3.4)	5 (1.5)	1 (2.6)	0 (0)
Undergraduate, *n* (%)	214 (80.1)	281 (85.2)	-	-
Year of degree *n* (%)				
First	113 (42.3)	123 (37.3)	-	-
Second	79 (29.6)	120 (36.4)	-	-
Third	58 (21.7)	63 (19.1)	-	-
Fourth	17 (6.4)	24 (7.3)	-	-
Years since completing first health degree	-	-	22 (10)	20 (15, 24)
Trained health professional, yes *n* (%)	-	-	32 (84.2)	28 (93.3)
Health professional currently practicing, yes *n* (%)	-	-	16 (50)	15 (53.6)
Encouraging patients to have a more physically active lifestyle in the last month, often (>6 patients) *n* (%)	-	-	9 (56.3)	8 (53.4)
Practice type, private practice *n* (%)	-	-	9 (56.3)	8 (53.3)
Patients/wk, median (Q1, Q3)	-	-	9 (4, 22)	6 (3, 15)
Years in practice, mean (SD)	-	-	18.6 (10.5)	22.5 (8.5)
Hours worked/wk, median (Q1, Q3)	-	-	13 (5, 40)	6 (4, 28)
How physically active do you think you are currently compared with other Australians of your sex and age? More active *n* (%)	140 (53.4)	159 (48.9)	18 (47.4)	20 (69)

* *p* < 0.01; T1: 2020; T2: 2021.

**Table 2 ijerph-19-09255-t002:** Health students and staff knowledge, role perception, confidence, feasibility, and barriers in relation to physical activity promotion pre- and post-intervention.

	Students				Staff			
Variable	Agree *n* (%)		Median (Q1, Q3)		Agree *n* (%)		Median (Q1, Q3)	
	T1	T2	T1	T2	T1	T2	T1	T2
Knowledge of physical activity messages ^a^								
Taking the stairs at work and generally being more active each day is enough physical activity to improve health	146 (49)	194 (54)	3 (2, 4)	2 (2, 4)	16 (37)	13 (41)	4 (2, 4)	4 (2, 4)
Half an hour of walking on most days is all the physical activity that is needed for good health	145 (49)	153 (42)	3 (2, 4)	3 (2, 4)	19 (44)	11 (34)	3 (2, 4)	4 (2, 4)
Physical activity that is good for health must make you puff and pant	87 (29)	90 (25)	4 (2, 4)	4 (3, 4)	15 (35)	13 (41)	4 (2, 4)	4 (2, 4)
Several short walks on most days is better than one round of golf per week for good health	231 (77)	285 (79)	2 (2, 2)	2 (2, 2) *	30 (70)	31 (97)	2 (2, 3)	2 (1, 2) *
Health Professionals role ^b^								
Discussing the benefits of a physically active lifestyle with patients/individuals is part of the health professional’s role	284 (95)	346 (95)	1 (1, 2)	1 (1, 2)	40 (93)	31 (97)	1 (1, 2)	1 (1, 2)
Suggesting to patients/individuals ways to increase daily physical activity is part of the health professional’s role	274 (92)	329 (91)	2 (1, 2)	2 (1, 2)	37 (86)	29 (91)	2 (1, 2)	1 (1, 2)
Health professionals should be physically active to act as a role model for their patients/individuals	268 (90)	324 (89)	2 (1, 2)	2 (1, 2)	38 (88)	26 (81)	2 (1, 2)	2 (1, 2)
Confidence in providing physical activity messages ^c^								
I would feel confident in giving general advice to patients/individuals on a physically active lifestyle	226 (76)	264 (73)	2 (1, 2)	2 (1, 3)	26 (70)	24 (83)	2 (1, 3)	2 (1, 2)
I would feel confident in suggesting specific physical activity programs for my patients/individuals	143 (48)	185 (51)	3 (2, 4)	2 (2, 4)	15 (42)	18 (62)	3 (1, 4)	2 (1, 4)
Barriers to physical activity promotion ^d^								
Lack of time	142 (50)	190 (55)	3.5 (3, 4)	4 (3, 4)	22 (56)	20 (67)	4 (3, 4)	4 (3, 4)
Lack of counselling skills	102 (36)	129 (37)	3 (2, 4)	3 (3, 4)	19 (49)	17 (57)	3 (3, 4)	4 (3, 4)
Lack of remuneration for promoting physical activity	65 (23)	63 (18)	3 (2, 3)	3 (2, 3)	11 (28)	3 (10)	2 (2, 4)	2 (1, 3)
Lack of interest in promoting physical activity	78 (28)	97 (28)	3 (2, 4)	3 (2, 4)	10 (26)	6 (20)	3 (2, 4)	3 (2, 3)
Feeling it would not change the patient’s/individual’s behaviour	118 (42)	129 (37)	3 (2, 4)	3 (2, 4)	12 (31)	13 (43)	3 (3, 4)	3 (3, 4)
Feeling it would not be beneficial for the patient/individual	44 (16)	38 (26)	2 (2, 3)	2 (2, 3)	3 (8)	4 (13)	2 (2, 3)	2 (2, 3)
Feasibility of physical activity promotion strategies ^e^								
Brief counselling integrated into regular consultations	224 (83)	259 (79)	2 (1, 2)	2 (1, 2)	19 (76)	26 (87)	2 (1, 2)	2 (1, 2)
Separate one-on-one consultations	162 (60)	181 (55)	2 (2, 3)	2 (2,4)	16 (42)	13 (43)	3 (2, 4)	3 (2, 4)
Group sessions	173 (64)	212 (64)	2 (1, 3)	2 (2, 3)	17 (45)	23 (77)	3 (2, 4)	2 (2, 2)
Distribution of resources (e.g., brochures)	229 (85)	285 (86)	1 (1, 2)	1 (1, 2)	33 (87)	27 (90)	1 (1,2)	1 (1, 2)

* *p* ≤ 0.05; T1: 2020; T2: 2021; ^a^ Student T1 *n* = 299 and T2 *n* = 363, Staff T1 *n* = 43 and T2 *n* = 32; ^b^ Student T1 *n* = 299 and T2 *n* = 363, Staff T1 *n* = 43 and T2 *n* = 32; ^c^ Student T1 *n* = 299 and T2 *n* = 366, Staff T1 *n* = 37 and T2 *n* = 29; ^d^ Student T1 *n* = 284 and T2 *n* = 345, Staff T1 *n* = 39 and T2 *n* = 30; ^e^ Student T1 *n* = 271 and T2 *n* = 330, Staff T1 *n* = 38 and T2 *n* = 30.

**Table 3 ijerph-19-09255-t003:** Themes relating to barriers to the promotion of physical activity with supporting verbatim quotes.

Themes	Students	Staff
Lack of PA promotion skills	*Lack of confidence to counsel on physical activity especially with overweight or obese patients/individuals* (Public Health) *Not wanting to cause offence to a patient* (Physiotherapy)	*Lacks courage to discuss tough issues* (Nursing) *Not having skills in assessing ambivalence and motivation* (Nursing)
Feeling hypocritical	*Embarrassed by own lack of personal physical fitness* (Nursing) *Clinician’s own inactive lifestyle may impact their willingness to encourage clients to understand the benefits of physical activity* (Psychology)	*Not wanting to give advice on something they don’t do personally* (Physiotherapy)
Understanding the benefits of PA promotion by all HPs, for all patients, in all settings	*Focus on immediate issue* (Pharmacy) *Feeling it is not part of our role* (Speech Pathology) *Not wanting to counter/contradict gp/specialist recommendations* (Occupational Therapy)	*Difficulty with talking about physical activity in the setting as may not directly relate to the service we provide* (Vision Science & Optometry)
Organisational support is key	*Not promoted by the organisation health professionals work in* (Nursing)	*Lack of organisational support and reinforcement of fundamental care activities* (Nursing)

**Table 4 ijerph-19-09255-t004:** Characteristics of subjects containing physical activity content pre- and post-intervention.

Subject Characteristics	T1	T2
	(*n* = 41)	(*n* = 48)
Discipline, *n* (%)		
Occupational Therapy	3 (7.3)	4 (8.3)
Physiotherapy	12 (29.3)	12 (25)
Pharmacy	2 (4.9)	4 (8.3)
Nutrition and Dietetics	2 (4.9)	2 (4.2)
Nursing	2 (4.9)	3 (6.3)
Midwifery	7 (17.1)	7 (14.6)
Psychology	0 (0)	0 (0)
Vision Science and Optometry	0 (0)	1 (2.1)
Speech Pathology	0 (0)	0 (0)
Counselling	0 (0)	0 (0)
Sport and Exercise Science	7 (17.1)	6 (12.5)
Medical Radiation Science	0 (0)	0 (0)
Public Health	0 (0)	0 (0)
Inter-professional	6 (14.6)	9 (18.8)
Undergraduate, *n* (%)	24 (58.5)	30 (62.5)
Year of Degree, *n* (%)		
First	17 (41.5)	18 (37.5)
Second	8 (19.5)	12 (25)
Third	13 (31.7)	14 (29.2)
Fourth	1 (2.4)	2 (4.2)
Multiple years	2 (4.9)	2 (4.2)
Teaching Period, *n* (%)		
Semester 1	21 (51.2)	27 (56.3)
Semester 2	20 (48.8)	20 (41.7)
Winter Term	0 (0)	1 (2.1)
Number of Students/subject, median (Q1, Q3)	33 (25, 60)	41 (22, 65)
Specific physical activity content in lectures, yes *n* (%)	33 (80.5)	38 (79.2)
Number of lectures/subject, median (Q1, Q3)	2 (1, 3.5)	2 (1, 5)
Specific physical activity content in lectures (hrs), median (Q1, Q3)	0.5 (0.15, 1)	1 (0.15, 2)
Specific physical activity content in tutorials, yes *n* (%)	29 (70.7)	37 (77.1)
Number of tutorials/subject, median (Q1, Q3)	1 (0, 2.5)	1 (1, 5)
Specific physical activity content in tutorials (hrs), median (Q1, Q3)	0.5 (0, 1.75)	1 (0.35, 2.5)
Assessment of physical activity content, yes *n* (%)	21 (51.2)	28 (58.3)
Type of physical activity content included, yes *n* (%)		
Physical activity guideline components	22 (53.7)	29 (60.4)
Health benefits of physical activity	40 (97.6)	47 (97.9)
Physical activity promotion skills	26 (63.4)	36 (75)
Movement for Movement resources	2 (4.9)	14 (29.2) *

* *p* < 0.01; T1: 2020; T2: 2021.

## Data Availability

The datasets used and analysed during the study are available from the corresponding author upon reasonable request.

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
