# Peer review of "Australian University Nursing and Allied Health Students’ and Staff Physical Activity Promotion Preparedness and Knowledge: A Pre-Post Study Using an Educational Intervention"

_ijerph, 2022, doi:10.3390/ijerph19159255_

Round 1

Reviewer 1 Report

This is an article in the area of higher education curriculum, health and physical activity promotion. The article is well written with no grammatical errors . Whilst increasing PA levels is an important component to health care and preventative health esepcially, it is unclear as to what this article adds to the available literature on (1) equipping health professionals with an appropriate level knowledge and skills of exercise and provision of PA promotion strategies (2) the understanding of content in university degree programs and (3) the level of understanding of PA and exercise strategies in emerging health care practice professionals. The most important finding is the wide variation of understanding of PA requirements but most felt equipped to offer advice. In addition those perceiving their own level of inactivity of physical appearance felt less inclined to provide advice on PA.   Understanding PA recommendations is one element that should be understood but clearly ‘behavior change modifications’ and skills in health and lifestyle counselling /self management is more important to elevating PA levels but this is only briefly mentioned (page 6) . Eliminating the investigation of a key requirement / skill of emerging health professionals involved in PA promotion is a limitation. It is interesting that Exercise and Sport Science Australia who provide comprehensive guidelines on specific content for graduates of university programs in exercise and health is not discussed. Graduates in ‘exercise science ‘ , which are trained professionals providing PA advice and (2) ‘accredited exercise physiologists' who are active in the primary and secondary prevention space at not even mentioned / examined but rather a low number of participants in ‘sport and exercise science (n= approx 6% of cohort) It is confusing this study only examine 1 university thereby promoting inherent bias in results.

Reviewer 2 Report

The article is eligible for publication.

Reviewer 3 Report

Dear authors,

Congratulations on a study that is certainly very relevant for healthcare academic curricula.

I have some comments and suggestions to increase your manuscript's quality.

Title

- Your title isn't clear about the core topic of the study. It might sound like it's about the future nursing workforce having more PA itself or learning how to promote PA routines. I suggest you use your main keywords in the title, namely "nursing students", "PA promotion, preparedness and knowledge" and "PA resources".

Abstract

- You should mention the aims of the study in the abstract, which should be the same as you mention at the end of the Introduction. 

Introduction

- Why mention COVID-19 at the beginning of your Introduction (line 49)? Your study has nothing to do with COVID-19. 

- I suggest you restructure your Introduction. Since your study is about promoting awareness among healthcare students and future nurses about practices for PA promotion, your introduction should be directed to the importance of this topic, rather than philosophizing about how insufficient PA levels might be a risk factor for chronic diseases, the prevalence of PA among populations mentioned by WHO, etc, etc.

- I think you can find more recent studies regarding PA promotion for nursing and allied health students (lines 74-75).

- Wouldn't your first aim be "assess or evaluate PA promotion ..." rather than "describe PA promotion"?

Materials and Methods

- Information about informed consent and recruitment periods is missing.

- I understand that recruitment was performed using advertisements on an online study platform and via email. But, it's not clear how participants expressed interest in participating and contacted the research team. Don't forget that your study must be replicable in other contexts and, if this kind of information isn't clear, that goal can't be met.

- The criteria "if they self-reported they were a student in the Faculty and were planning a career as health professional" is somehow too general and subjective. How do you control this? 

- From 2.2 Intervention onwards, it's not clear what are the main outcomes of interest and how are they measured. Also, authors have to consider describing details to allow replication, including how and when interventions and/or outcomes were administered or measured.

- In 2.2.2, it's not clear what is a "PA champion". Further explanations would be welcome.

- Regarding the 2.3 Survey, the questionnaire that was used was adapted and validated for the Australian population? Could this information be included in this section?

Results

- It would be interesting, and easier for the reader, if you include baseline data in a table column, rather than describing it in the text.

Discussion

- There's a typo on line 435. It should read "COVID-19 pandemic may have affected implementation" rather than "effected implementation".

References

- 30% of your references have 10 or more years. If possible, please update some of them.

Round 2

Reviewer 3 Report

Dear authors,

Congratulations on the review and thank you for considering some of my suggestions.

Although not an issue that influences the publication of your manuscript, which I'm now recommending for publication with no further reviews, I still don't understand why you mention the COVID-19 pandemic. There's been a recent trend in scientific papers to systematically include COVID-19 as a mandatory justification for the importance of conducting all kinds of studies and a false motivation from authors that, performing that inclusion, journals will automatically consider the paper as very important for the scientific community. This is not true and can't be encouraged by research teams, reviewers or journals. 

Again, congratulations and good luck with your research!